# COVID-19: Present and Future Challenges for Dental Practice

**DOI:** 10.3390/ijerph17093151

**Published:** 2020-04-30

**Authors:** Najla Dar-Odeh, Hamzah Babkair, Shaden Abu-Hammad, Sary Borzangy, Abdalla Abu-Hammad, Osama Abu-Hammad

**Affiliations:** 1College of Dentistry, Taibah University, Al Madinah Al Munawara 43353, Saudi Arabia; Hbabkair@taibahu.edu.sa (H.B.); sbarzanji@taibahu.edu.sa (S.B.); oabuhammad@taibahu.edu.sa (O.A.-H.); 2School of Dentistry, University of Jordan, Amman 11942, Jordan; Shadenabuhammad@gmail.com; 3School of Medicine, University of Jordan, Amman 11942, Jordan; Abdullah018ju@gmail.com

**Keywords:** COVID-19, dentistry, oral, manifestations, saliva, aerosol

## Abstract

COVID-19 was declared a pandemic by the World Health Organization, with a high fatality rate that may reach 8%. The disease is caused by SARS-CoV-2 which is one of the coronaviruses. Realizing the severity of outcomes associated with this disease and its high rate of transmission, dentists were instructed by regulatory authorities, such as the American Dental Association, to stop providing treatment to dental patients except those who have emergency complaints. This was mainly for protection of dental healthcare personnel, their families, contacts, and their patients from the transmission of virus, and also to preserve the much-needed supplies of personal protective equipment (PPE). Dentists at all times should competently follow cross-infection control protocols, but particularly during this critical time, they should do their best to decide on the emergency cases that are indicated for dental treatment. Dentists should also be updated on how this pandemic is related to their profession in order to be well oriented and prepared. This overview will address several issues concerned with the COVID-19 pandemic that directly relate to dental practice in terms of prevention, treatment, and orofacial clinical manifestations.

## 1. Introduction

COVID-19 was declared a pandemic by the World Health Organization (WHO), with substantial numbers of infected cases and deaths reported in many countries. Among these countries, Italy, the United Kingdom, and Spain had a high fatality rate ranging 4–8% [1]. The disease is caused by one of the coronaviruses, which are a large family of viruses that may cause severe illnesses, such as Severe Acute Respiratory Syndrome (SARS) and Middle East Respiratory Syndrome (MERS). These viruses are common in animals with the potential of transmission to humans. They are composed of an envelope, a lipid layer, and single-stranded large RNA. The name “corona” (“crown” in latin) is attributed to the spherical shape and surface projections. Four subfamilies have been identified: alpha-, beta-, gamma-, and delta-coronaviruses. Beta-coronaviruses seem to originate from mammals, namely bats; it was found that the genome sequence of SARS-CoV-2, the virus responsible for COVID-19, is >90% identical to a bat coronavirus RaTG13. In fact, bats represent a natural reservoir for a wide variety of coronaviruses including SARS-CoV-like and MERS-CoV-like viruses. SARS-CoV-2 is closely related to the SARS-CoV virus, and it belongs to the B lineage of the beta-coronaviruses, which are known to cause severe disease and fatalities. 

The origin of SARS-CoV-2 is still unclear, however, initial transmission from animals to humans has probably occurred in the Huanan seafood market in Wuhan, China in December 2019, where a cluster of pneumonia cases, caused by a newly identified β-coronavirus, were diagnosed in this city. 

The main clinical manifestations are respiratory in nature, and they manifest after a mean incubation period of five days (range: 0–24 days). An increased risk of infection was found in patients with certain co-morbidities mainly including hypertension, diabetes, and ischemic heart disease. A possible explanation for this association could relate to the nature of these diseases and the types of medications used for treatment. In hypertensive and diabetic patients, circulating amounts of angiotensin converting enzyme-2 (ACE2) are increased. Moreover, some drugs including some types of antihypertensive drugs act as ACE inhibitors which further increase ACE2, and as SARS-CoV-2 binds to the host cell’s membrane via ACE2, an increased risk to infection is noticed [2]. There is a wide variation between countries in the numbers of deaths and positive asymptomatic cases, with some reports indicating that approximately 80% of infected cases are asymptomatic [3]. Initial symptoms consist of fever, cough, nasal congestion, fatigue and other signs of upper respiratory tract infections. In approximately two thirds of the cases, the infection can progress to severe disease with dyspnea and severe lung congestion. Multi-organ failure may eventually result in the form of respiratory failure, shock, acute respiratory distress syndrome, arrhythmia, acute myocardial injury, acute liver injury, and sepsis [4].

As of 26 April 2020, there have been more than 2.9 million cases, and more than 205,000 deaths globally. In response to this challenging pandemic, the Center for Disease Control and Prevention (CDC), American Dental association (ADA), the National Health Service (NHS), as well as other health regulatory bodies have provided advice to dentists to regulate dental services and to provide them with guidance in order to protect themselves, their co-workers, and their patients from this infection.

Dentists are among the highest risk categories for transmission and contraction of the coronavirus, with many routine dental procedures having the potential to transmit the virus through aerosols. Asymptomatic (carrier) patients as well as patients with an acute respiratory illness may present for dental treatment at outpatient dental settings. While it is important to provide treatment for patients who present with urgent or emergency dental procedures, the primary goal should be to prevent transmission of infection to patients and dental healthcare personnel. The growing fear of cross-infection, and the possible role of dental practice in spreading the infection, have obliged dentists to step aside and to confine themselves in home quarantine similar to other non-healthcare sectors of the population. In addition, there has been an increased demand for personal protective equipment (PPE), which consists of garments to protect healthcare workers or any other persons to get infected. The standard PPE consist of gloves, mask, and gown. However, in case of airborne infections like COVID-19, additional equipment should be utilized including face protection, goggles, mask, face shield, gloves, gown or coverall, head cover, and rubber boots [5].

Dentists are now providing emergency dental procedures only, during which they have to follow the recommended cross-infection control protocols. Therefore, the main bulk of published research directed to dentists has mainly focused on giving a background on the pandemic and what the recommended cross-infection control measures are.

There are many aspects of COVID-19 that are related to dental practice in addition to infection control, including prevention and treatment. There are also a number of clinical manifestations that affect the orofacial region and that dentists should be familiar with. This overview addresses the medical-dental aspects of COVID-19 infection. It is directed to dental healthcare personnel to update them on the recommended guidelines for provision of dental health services during this critical period, and to explain important aspects of the COVID-19 infection with relevance to the orofacial region and oral healthcare. These aspects are divided into three sections: prevention, treatment, and oral manifestations.

## 2. Prevention: Provision of Dental Healthcare during the COVID-19 Pandemic

The ADA has maintained a consistent stand since the pandemic was recognized. They called upon dentists to postpone elective dental procedures for all dental patients, and to provide dental treatment only for urgent or emergency cases [6]. The main aims were to minimize potential for healthcare transmission of COVID-19, and to avoid shortage of PPE for healthcare personnel caring for those with COVID-19, or dental healthcare personnel providing urgent dental care in emergency cases. They further clarified the meaning of dental emergencies as “potentially life-threatening conditions that require immediate treatment to stop ongoing tissue bleeding, alleviate severe pain, or infection”; therefore, the emergency conditions indicated for treatment include cellulitis, uncontrolled bleeding, or trauma [7]. Within this context, provision of urgent dental treatment is to be done in regular dental clinics, and not to direct patients to emergency rooms even afterhours unless a life-threatening emergency is encountered. Types of urgent dental care was also clarified in detail to include: severe dental pain; certain infections such as pericoronitis, postoperative osteitis, dry socket, or abscess/cellulitis; trauma such as symptomatic fractured tooth or avulsion/luxation; as well as certain urgent restorative procedures [7].

The NHS, on the other hand, has initially provided advice to dentists to perform routine dental care only for patients with no symptoms of COVID-19, provided that no aerosol-generating procedures are undertaken. Recommendations of the NHS were updated according to the evolving situation of the pandemic so that the most recent recommendation was in concordance with that of the ADA.

## 3. Treatment: Dental Aspects of Drugs Used for COVID-19 Treatment

So far no definitive treatment is adopted for COVID-19. A number of antiviral drugs as well as other drug categories were used so far with variable success rates. Some of these drugs have direct relevance to dental practice including analgesics, hydroxychloroquine, and azithromycin.

### 3.1. Azithromycin

Azithromycin is a macrolide antibiotic that is particularly important in dental practice. It is a recommended antibiotic in the empiric treatment of odontogenic infections mainly in penicillin-allergic patients [8]. It is also among the top five antibiotics prescribed in the dental setting in some countries including the USA, Brazil, and Belgium [9,10,11]. The long half-life of azithromycin make it a favorable antibiotic for children who lack compliance and for whom a once daily oral dosage is recommended. Further, it is effective in the management of respiratory infections in young children [12].

Hospitalized patients usually receive the intravenous form of the drug for the treatment of community-acquired pneumonia. This antibiotic is considered relatively safe in adults, children, and pregnant women [13]. However, a number of side effects have been identified especially with intravenous administration, which may be associated with gastrointestinal disturbances, ototoxicity, and pain and inflammation of the injection site [14] The development of resistant bacteria, [15] and its association with proarrhythmic events [16] have also been reported. The latter risk has been attributed to QT prolongation (summation of action potential of ventricular myocytes), which can lead to a life-threatening arrhythmia; however, susceptible patients usually have other co-factors such as old age, heart disease, and exposure to other QT prolonging drugs [17].

In vitro studies have shown that azithromycin is active against Zika and Ebola viruses, [18,19,20] and is able to prevent severe respiratory tract infections when administrated to patients suffering viral infection [12] However, the efficacy of azithromycin in combination with hydroxychloroquine in the treatment of COVID-19 patients has not been confirmed yet [21,22], and more studies are needed to further investigate its clinical effects. In light of the current shift of dental services towards the provision of emergency treatment only, and the possible increase in antibiotic prescriptions for severe orofacial infections, the use of azithromycin in dentistry should be monitored, especially that its use in dental practice as a favorable antibiotic is reported in countries with a high toll of COVID-19 infections. Alternative antibiotics such as amoxicillin or clindamycin (in penicillin-allergic patients) should be considered for indicated cases, provided that no contraindications are present. An important example is patients who has a history of pseudomembranous colitis or ulcerative colitis, and hence cannot use clindamycin [23]. Dentists and physicians working in the treatment of emergency dental cases should be vigilant in prescribing antibiotics only for indicated cases and should consider the use of analgesic alternatives to control dental pain. Avoiding the development of side effects and antibiotic resistance should be considered among the goals of treatment. 

### 3.2. Chloroquine

Chloroquine is an antiparasitic drug that is primarily used as antimalarial drug since the 1930s. It has recently attracted a lot of attention due to its use in the treatment of COVID-19. However, its use in the treatment of some oral diseases has been recognized for a long time. It was noticed to possess efficacy towards autoimmune diseases and has been implemented since the 1980s in the treatment of systemic lupus erythematosus (SLE), a disease that may have oral manifestations like ulcers. Its use in the treatment of primary Sjögren’s syndrome has been suggested by some scientists [24], and it is also recommended for the treatment of chronic ulcerative stomatitis [25]. It had been suggested for the treatment of oral squamous cell carcinoma due to its role in cell protection by eliminating excessive proteins and injured/aged organelles in the microenvironment of tumors with subsequent acceleration of tumor cell death [26].

The antiviral activity of the drug has long been recognized. In the current epidemic of COVID-19 many countries announced its use in their trials to eradicate this disease. Scientists stated that the drug, which has established antiviral activity over the past 40 years, inhibited SARS-CoV-2 viral replication in vitro and human clinical application indicated apparent efficacy [27].

Hydroxychloroquine is a derivative of chloroquine with significantly higher solubility, and lower toxicity, therefore fewer side effects are anticipated [28].

Pharmacological modelling based on observed drug concentrations and in vitro drug testing suggest that prophylaxis with hydroxychloroquine at approved doses could prevent SARS-CoV-2 infection and ameliorate viral shedding [29]. The combination of antiviral drugs, such as remdesivir and chloroquine, has been considered highly effective in the control of infection in vitro and has been suggested in the treatment of Covid-19 due to its safety profile [30]. However, clinical trials conducted so far are limited in sample size and their lack of randomization cast doubt on reported outcomes.

It is still unknown how this drug exerts its anti-viral activity, but some researchers believe it can inhibit the development of an acidic media in endosomes that transport it from the cell membrane to cytoplasm. Alkaline media in endosomes is believed to prevent viral transfer to cytoplasm and can thus limit the replication of several viruses [31]. The activity of the drug against autoimmune diseases, such as SLE, is believed to be due to its action to prevent production or release of IL-6 and TNF- α, and due to its inhibitory action on autophagy [32]. This activity of hydroxychloroquine has been demonstrated in vitro against influenza and coronaviruses, however, clinically in humans and on animals the therapeutic activity was less successful.

The drug is generally safe, with poisoning being associated with the dangerous side-effects of retinopathy and immunosuppression [33]. However, it is contraindicated in pregnancy. During the current pandemic of COVID-19, and due to increased demand, severe shortages of the drug were reported and adversely affecting on the regular autoimmune disease patients with countries banning its export. Dentists have to be aware that shortages of chloroquine may influence their patients who are dependent on this drug especially SLE and Sjogren’s syndrome patients who have oral manifestations. They also should be aware of the possible oral complications caused by the drug, namely melanotic pigmentation of the oral mucosa [34] and lichenoid reaction [35].

### 3.3. Ibuprofen and NSAIDS in COVID-19 Patients

Since the recognition of the COVID-19 pandemic, professional regulatory bodies advised against provision of dental treatment except for emergency cases. It became essential that dental patients will rely on supportive therapy such as analgesics, and non-steroidal anti-inflammatory drugs (NSAIDs) for the control of dental symptoms of pain. Among these patients, some may be asymptomatic for COVID-19. Furthermore, supportive analgesic, antipyretic therapy remains the backbone for the treatment of mild to moderate cases of COVID-19. This may eventually lead to the increased demand on analgesics. There was a warning against the use of ibuprofen in the treatment of COVID-19 due to the increased expression of angiotensin-converting enzyme-2, which is believed to be the binding receptor of the virus to the cells. Consequently, the accelerated expression of this protein would theoretically potentiate and enhance the infection.

This argument may be based on mechanistic or theoretical pharmacology rather than evidence-based clinical trials [36]. COVID-19 can be such a severe infection in about 20% of the cases, forcing patients to choose ibuprofen as a more effective drug compared to paracetamol. Thus, the clinical manifestations potentially emanate from this fierce infection itself rather than the theoretical potentiating action of the drug. Further, there is no strong epidemiological evidence to suggest a harmful effect of ibuprofen on COVID-19 patients [36].

The WHO recommendation in this case is to use paracetamol as first line treatment, while ibuprofen comes as second line treatment [37]. Recently, the National Institute for Health and Care Excellence (NICE) said that there is no evidence from published scientific studies to determine whether acute use of NSAIDs is related to increased risk of developing COVID-19 or increased risk of a more severe illness [38]. This was confirmed by the NHS England in their recent commissioning policy for acute use of NSAIDs for people with or at risk of COVID-19 [39].

Dentists should remain updated as more information emerges on the topic and should weigh any benefits against harm when prescribing analgesics for patients with dental pain. Paracetamol can be used as a first line analgesic, however, if it is not effective, they can prescribe ibuprofen or other NSAIDs unless there is a contraindication. 

## 4. Oral Manifestations of COVID-19

### 4.1. Salivary Gland Infection

The genome of COVID-19 virus has been detected in saliva in the majority of patients with this disease [40], indicating the potential infection of salivary glands [41]. It is interesting to know that in some cases, COVID-19 was only detected in saliva, with no evidence for its presence in the nasopharynx [42].

Positive salivary tests indicate possibility of transmission through the spread of saliva as respiratory viruses usually spread via direct contact or spatter and aerosol production from mouth and nose i.e., sneezing or coughing [41]. Furthermore, respiratory droplets containing influenza virus have been detected even during normal breathing [43].

Although it is possible to detect the virus in saliva with viral culture, this should be interpreted with caution since saliva may contain secretions that originate from the nasopharynx or the lungs through the action of cilia [41]. However, this can be ascertained by choosing the correct method of saliva collection; to collect saliva from a particular salivary gland (the parotid gland for example) rather than obtaining the sample directly from the mouth [44].

The detection of virus in saliva is being used for monitoring saliva virus load during serial viral load monitoring instead of nasopharyngeal or oropharyngeal sources to reduce patient discomfort and health hazards to the operator during successive sampling [41]. Conducting COVID-19 tests on saliva is easier for the patient and operator and the process bears less risk of cross contamination. Specimens can be provided by asking the patient to spit into a sterile container, and the operator stands little chance of exposure with such a non-invasive procedure [41]. Saliva collection is more comfortable for patients than venipuncture as well as being more cost-effective with minimal required instruments [45].

This finding is of particular interest to dentists. The initial recommendation by the NHS was to provide treatment to all patients except those with symptoms of infection. Also, all dental treatment was allowed except procedures that are associated with aerosolization. However, it is established now that there is a proportion of asymptomatic patients who may transmit infection, and the presence of the virus in saliva means that even non-aerosol producing dental procedures can be a source of infection. Another important aspect of this finding is that dentists who are engaged in tobacco cessation efforts should disseminate awareness among their smoker patients of the possibility of salivary virus transmission via social sharing of tobacco smoking instruments namely the electronic cigarettes and waterpipe [46].

### 4.2. Taste Abnormalities

Loss of taste and smell have been recognized lately as one of the symptoms of COVID-19 [47]. An Italian team reported that 20 out of 59 COVID-19 patients who were interviewed (33.9%) had at least one taste or olfactory disorder and 11 (18.6%) had both [47]. Most of the patients with these symptoms (91%) reported the occurrence of taste alterations before being hospitalized. Taste and smell disorder in this case could be explained by the fact that SARS-CoV-2 has been known for its interaction with angiotensin converting enzyme 2 (ACE2) receptor, to facilitate its penetration into the cell, and this receptor is widely expressed on the epithelial cells of oral mucosa and the brain [48]. In fact, expression of ACE2 was found to be higher in tongue, where the taste buds are most abundant, than gingiva or buccal mucosa [48]. Another possibility is that SARS-CoV-2 could also be detected in saliva and infection of salivary glands is also possible [40], which increases the availability of virus in the oral cavity and its uptake by the epithelial cells.

Dentists should be aware of this symptom since they may encounter patients with taste abnormalities in the form of dysgeusia or burning mouth syndrome. This is particularly important because these symptoms may precede the onset of respiratory diagnostic manifestations of the disease. However, reporting of this symptom should be interpreted with caution as the affected patients are known to be of the old age group who are already susceptible to taste and smell disorders.

## 5. Discussion and Recommendations

Dental practitioners have an important role in the global fight against pandemics like COVID-19. They are experienced in cross-infection control procedures and barrier techniques. They are competent in suture placement, hemostatic procedures, and in many countries, they can perform parenteral drug administration. They are also constructive members in multidisciplinary professional groups and experienced in managing patients in pain. They are well adapted to management of vulnerable patients including children, pregnant women, and elderly people. On the other hand, dentists should recognize the importance of following the regulations for delivery of oral healthcare so as to protect their patients and members of the dental healthcare team.

Practicing dentists should ensure that all members of the oral healthcare team are well acquainted to the COVID-19 transmission and preventive measures. Provision of dental services should take into consideration the availability of PPE, and that only emergency cases are admitted for treatment. Cross-infection control measures should be applied meticulously at all times, and social distancing should be adopted in the practice unless advised otherwise. Dental procedures in general are categorized into two groups according to aerosol generation. Most dental procedures generate aerosol; preparing cavities for fillings, use of rotary instruments for root canal treatment, scaling and polishing of teeth, dental implantation, and surgical removal of teeth are only some examples. Asymptomatic COVID-19 patients may present for emergency dental treatment. These patients are expected to have saliva contaminated with the virus and they are a confirmed source of infection. Moreover, the conjunctiva mucosa and upper respiratory tract are connected by the nasolacrimal duct, and they share ACE2 on the cell membrane [49]. This exposes dental healthcare personnel to the risk of infection via direct exposure of conjunctiva (eyes) to droplets from patients during dental treatment.

There are now restrictions for the work of dentists in many countries, however, some countries like Austria and Jordan will start to ease the lock down. In Jordan, for instance, dentists will be permitted to work in their practices as of 27 April 2020, even though complete eradication of the virus has not been accomplished yet. Considering that neither treatment nor vaccination is available for COVID-19, it would be wise for dentists to rely more on non-aerosol generating procedures for treatment of their patients. Excavation of caries rather than drilling and conventional root canal treatment rather than rotary instruments, for example, should be the mainstay of treatment at this point of time. Researchers should focus on developing barrier techniques and negative pressure procedures to contain and isolate the aerosol so that dental procedures are safe for dental healthcare personnel and patients alike.

Furthermore, dentists should be aware that COVID-19 patients may present with oral symptoms that are suspected to be linked to the virus such as taste abnormalities.

While home confinement is considered the mainstay for populations to prevent transmission of the virus, dentists should not be confined by the society within the borders of their specialty. They have many roles to play. Following the recommended cross-infection control procedures, spreading awareness based on evidence and not misconceptions, identifying emergency cases indicated for dental treatment, and practicing effective tele-dentistry when needed can all be helpful for dental patients and community as a whole. Dentists should give drug prescription particular attention. When advising patients to use medications for treatment of dental problems it is important to consider the indicated clinical conditions for analgesics, anti-inflammatory drugs, and antibiotics, and patients should be advised against using antibiotic self-medication to relieve dental pain. 

It is certainly the right time for dental schools to expand the learning outcomes of their courses to include additional roles of dentistry that take into consideration natural disasters and pandemics. Furthermore, dentists should be prepared to be active members in healthcare teams dealing with pandemics. Professional dental associations should contemplate continuing educational courses for practicing dentists that reinforce their role in the healthcare team by delivering courses on essential aspects of acute healthcare such as basic life support, phlebotomy, and drug prescribing.

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
