# Peer review of "COVID-19: Present and Future Challenges for Dental Practice"

_ijerph, 2020, doi:10.3390/ijerph17093151_

Round 1
Reviewer 1 Report
The manuscript is very interesting. I just have two questions:
The affirmation on lines 154 and 155 need the reference;
Please cite the health organisations referred on lines 208 and 209.
Author Response
Response to reviewer 1
Thank you for your valuable comments. All your comments were addressed as you can see below. Modifications in text were highlighted in yellow.
- English language and style are fine/minor spell check required
Response: text was revised for spelling mistakes
- The manuscript is very interesting. I just have two questions:
The affirmation on lines 154 and 155 need the reference;
Response: Thank you. A reference was cited (Ponticelli, C.; Moroni, G. Hydroxychloroquine in systemic lupus erythematosus (SLE). Expert Opin. Drug Saf. 2017) to support the safety of hydroxychloroquine and the possible consequences of poisoning: retinopathy and immunosuppression.
- Please cite the health organisations referred on lines 208 and 209.
Response: “Health authorities” was replaced by the “NHS” based on their initial recommendation early in the course of the pandemic.
Reviewer 2 Report
Dear Authors,
My suggestions of corrections are :
tittle ( line 3 ) : I think ideally profession should be changed to field
line 14 line 30 , line 40 : put some information about the medium death rate already reported in the literature
line 17 : give some examples of countries , explain the definition of dental emergencies in Covid 19 situation and give examples of clinical situations
line 18 : add family and contacts to the list of risk persons .
line 19 , 20 : Put ( PPE ) after personnel protective equipment .
The sentence regarding the PPE is not true in all the countries ( also in lines 84 , 85 ) . In some countries that they only have private practices ( ex Portugal ) , the private practices are the ones treating emergencies and for that they need the PPE in their offices.
line 26 : change oral by mouth
Line 31 : remove common cold because it creates confusion with influenza virus
line 33 : change for common in animals and sometimes transmitted to humans . Add lipídic layer
line 34 : crown add in Latin
line 40 : explain the symptoms start in oro faringe and respiratory tract before evolving for multi organ failure .
Line 48 : refer that about 80 % of the infected are asymptomatic
line 50 : error in writing dyspnea ; change chest subtis per lung congestion
line 53 : revise the latest world statistics
line 57 put after themselves , their workers
line 65 : put PPE definition here or in introduction/ abstract
line 83 : add emergency cases that couldn’t be postponed
line 100 : add after infections , namely in the penicillin allergic patients
line 101 : missing one , after USA
line 109 : explain the mechanism why azytromicine changes the heart rhythm ( elevates QT interval )
line 119 : put between () ( in penicillin allergic patients )
line 125 : say that chloroquine is an anti parasite drug
line 140 : explain why hidroxi chloroquine is more effective than chloroquine
line 142 : change it by of infection
line 161 : add and lichenoid reaction
line 165: add after analgesics : and NSAIDs
line 179 And 182: add use of paracetamol and ...
Add a 3.4 item explaining that anti hypertension medication that blocks the ACE2 can also interfere with COVID 19 infection and explain the mechanism
line 238 : not in all the countries , dentists can do parenteric drug administration
line 251 : add eyes after conjuntiva
line 270 : add Anti - inflammatory after analgesics ( pain killers )
Author Response
Thank you for your valuable comments comments. All your comments were addressed as you can see below. Modifications in text were highlighted in yellow.
- Moderate English changes required
Text was revised for English language
- tittle ( line 3 ) : I think ideally profession should be changed to field
Response: We appreciate this comment and your opinion is respected. We thought about changing the word “profession” and most of us thought that the word “practice “ is more appropriate so that the new title will be : COVID-19: Present and Future Challenges for Dental Practice.
- line 14 line 30 , line 40 : put some information about the medium death rate already reported in the literature
Response: Fatality rate of some countries were added: line 15, 31,32
- line 17 : give some examples of countries , explain the definition of dental emergencies in Covid 19 situation and give examples of clinical situations
Response: Line 17: “ The American Dental Association” was added.
Definition of dental emergencies , their types and types of urgent dental care according to the American dental association now appears in text lines 98-107
- line 18 : add family and contacts to the list of risk persons .
Response: family and contacts are added line 19
- line 19 , 20 : Put ( PPE ) after personnel protective equipment .
Response: PPE was added line 21
- The sentence regarding the PPE is not true in all the countries ( also in lines 84 , 85 ) . In some countries that they only have private practices ( ex Portugal ) , the private practices are the ones treating emergencies and for that they need the PPE in their offices.
Response: a sentence was added in line 98 to clarify that PPE is also needed for emergency dental situations.
- line 26 : change oral by mouth
Response: We completely understand the confusion that the word “oral” may cause, so we changed it to “orofacial” because this word denotes an anatomical area that is more accurate than “oral”. We hope that the honorable reviewer approves that.
- Line 31 : remove common cold because it creates confusion with influenza virus
Response: “ common cold” was removed
- line 33 : change for common in animals and sometimes transmitted to humans . Add lipídic layer
Response: the sentence now goes:” common in animals with the potential of transmission to humans” line 35
“lipid layer” was added in line 36
- line 34 : crown add in Latin
Response: “in latin” added line 36
- line 40 : explain the symptoms start in oro faringe and respiratory tract before evolving for multi organ failure .
Response: This was added to the third paragraph of introduction (line 59-61)
- Line 48 : refer that about 80 % of the infected are asymptomatic
Response: This was added with a reference (line 56)
- line 50 : error in writing dyspnea ; change chest subtis per lung congestion
Response: Dyspnea was corrected and lung congestion was placed in the sentence (line 59)
- line 53 : revise the latest world statistics
Response: statistics updated (April 26)- 1ine 63
- line 57 put after themselves , their workers
Response: The term “co-workers” was added after themselves-line 67
- line 65 : put PPE definition here or in introduction/ abstract
Response: PPE definition and a reference were added-line 76-80
- line 83 : add emergency cases that couldn’t be postponed
Response: Definition of dental emergencies , their types and types of urgent dental care according to the American dental association now appears in text lines 99-107.
- line 100 : add after infections , namely in the penicillin allergic patients
Response: This was added-line 121
- line 101 : missing one , after USA
Response: This was added-line 122
- line 109 : explain the mechanism why azytromicine changes the heart rhythm ( elevates QT interval )
Response: The QT prolonging effect of azithromycin was explained and a reference was cited (Choi, Y.; Lim, H.S.; Chung, D.; Choi, J.G.; Yoon, D. Risk evaluation of azithromycin-induced QT prolongation in real-world practice. Biomed Res. Int. 2018.)- 1ine 131-134
- line 119 : put between () ( in penicillin allergic patients )
Response: Added – line 143
- line 125 : say that chloroquine is an anti parasite drug
Response:Added- line-151
- line 140 : explain why hidroxi chloroquine is more effective than chloroquine
Response:
This was added with a reference- line 165-166
- line 142 : change it by of infection
Response: Modified Line 170
- line 161 : add and lichenoid reaction
Response: Modified- Line 189
- line 165: add after analgesics : and NSAIDs
Response: Modified, line-193
- line 179 And 182: add use of paracetamol and ...
Response: Acetaminophen is replaced by paracetamol throughout the text because among the two synonyms the term “paracetamol” seems to be used more commonly than acetaminophen worldwide.
- Add a 3.4 item explaining that anti hypertension medication that blocks the ACE2 can also interfere with COVID 19 infection and explain the mechanism
Response: This was explained in the third paragraph of introduction and a reference was added line 49-54
- line 238 : not in all the countries , dentists can do parenteric drug administration
Response: The sentence was modified taking this note in consideration (first paragraph of discussion line 270-271)
- line 251 : add eyes after conjuntiva
Response: “eyes” was added after conjunctiva line 288
- line 270 : add Anti - inflammatory after analgesics ( pain killers )
Response: Anti-inflammatory was added line 309
Reviewer 3 Report
Good overview but not strong on evidence based suggestions on how to alter provision of therapy.
1) Rationale for not using Azythromycin in dental infections is not clear to me. "Because it may help against virus" does not yield any evidence that should limit its use in dentistry. References for this would be helpful
2) As a practicing dentist the section on analgesics is not particularly helpful. Are all NSAIDS an issue with COVID or just ibuprofen? An evidence based review more than just what thw WHO says would be helpful. Original sources?
A strong evidence based summary section clearly outlining how dentists need to alter or at least adjust their clinical treatment would be most helpfu
Author Response
Thank you for your valuable comments. All your comments were addressed and modifications were highlighted in yellow.
- English language and style are fine/minor spell check required
Text was revised for English language
- Good overview but not strong on evidence based suggestions on how to alter provision of therapy.
Response: Thank you. Not all data on COVID-19 are clear and researchers worldwide are still in the process of collecting answers to important aspects of this pandemic. We did our best in this overview to refer to reliable websites and papers published in reputable journals. We are aware that much improvement should be done and we will consider updating the dental community and policy makers of any new information that may emerge.
We did revision of the original manuscript taking into consideration the most recent references published on the topic.
- 1) Rationale for not using Azythromycin in dental infections is not clear to me. "Because it may help against virus" does not yield any evidence that should limit its use in dentistry. References for this would be helpful
Response: The sentence: “the use of azithromycin in dentistry should be revised” was replaced by the sentence: “the use of azithromycin in dentistry should be monitored”- line 141. This is to alert dentists and health policy makers that the demand on azithromycin may increase, hence they should consider other effective alternatives like amoxicillin and clindamycin, unless there is a contraindication to these antibiotics. Advice on selection of antibiotics with a reference was given- line145-146
- 2) As a practicing dentist the section on analgesics is not particularly helpful. Are all NSAIDS an issue with COVID or just ibuprofen? An evidence based review more than just what thw WHO says would be helpful. Original sources?
Response: other resources were added to clarify this issue, and to include also NSAIDs: line 208-212
- A strong evidence based summary section clearly outlining how dentists need to alter or at least adjust their clinical treatment would be most helpful
Response: summary section has relied mainly on the updated information on COVID-19 and how it intersects with the dental profession. We added some more sentences to give the dental community more advice regarding provision of care: line 276-281
Round 2
Reviewer 3 Report
all concerns addressed
This manuscript is a resubmission of an earlier submission. The following is a list of the peer review reports and author responses from that submission.